# Large Uncertainty in Observed Estimates of Tropical Width From the Meridional Stream Function

**Daniel Baldassare[1], Thomas Reichler[1], Piret Plink-Björklund[2], and Jacob Slawson[2]**

- [1]Department of Atmospheric Sciences, University of Utah, Salt Lake City, UT 84112, USA
- [2]Department of Geology and Geological Engineering, Colorado School of Mines, Golden, CO 80401, USA

Correspondence: Daniel Baldassare (daniel.baldassare@utah.edu)

**Abstract**

Recent Hadley cell expansion rate estimates vary substantially, as a multitude of methods and reanalysis datasets yield conflicting results. Among the many methods of estimating the Hadley cell width, the meridional stream function 500 hPa zero-crossing is the most widely used, as it is directly related to the poleward edge of the Hadley Cell (HC). Other common metrics use atmospheric phenomena associated with the HC as a proxy, for instance the zonal surface wind zero-crossing. As each of these metrics require different reanalysis data, each with varying error, the level of data-driven uncertainty differs between each metric. While previous work has analyzed the statistical and dynamical relationships between metrics, to date no study has quantified and compared the uncertainty due to reanalysis data error in different HC metrics. In this study, we use ERA5 ensemble members, which include small perturbations in atmospheric variables based on the data error, to quantify the uncertainty associated with six commonly used HC metrics as well as the range of their trend estimates. In the Northern Hemisphere, the tropical expansion rate calculated by the stream function is roughly 0.05 degrees per decade, while the Southern Hemisphere rate is 0.2 degrees per decade over the period from 1979-2022. Of the six metrics, only the meridional stream function and precipitation minus evaporation have substantial uncertainties. The stream function errors are large due to uncertainty in the underlying meridional wind data and the

presence of large regions of near-neutral circulation at the poleward edge of the tropics. These errors have decreased in recent decades because of improvements in the assimilated observations. Despite these improvements, metrics which use well-observed and constrained quantities such as the zonal surface wind zero crossing have lower uncertainty, particularly in summer and fall in the Northern Hemisphere.


**Plain Text Summary**

Using ensemble members from the ERA5 reanalysis, the meridional stream function, the most widely used method for estimating trends in the extent of tropical circulation, was found to have large error, particularly in the Northern Hemisphere and in the summer, because of weak gradients at the tropical edge and poor data quality. Other methods using better observed data were found to have smaller error.

## 1    Introduction

Observed estimates of Hadley Cell (HC) meridional extent change in the last few decades vary substantially in the literature, as a variety of methods and reanalysis datasets, as well as time periods, yield conflicting results (Staten et al., 2018; Xian et al., 2021). HC expansion has been a well-studied phenomenon, as the poleward edge of the HC is associated with decreased precipitation, particularly over the ocean (Schmidt and Grise, 2017). During the 2000s, HC expansion rates were estimated across a wide range of positive values as high as 3 degrees per decade, while recent annual mean trend estimates have decreased to 0.2-0.4 degrees per decade (Grise et al., 2019). The strong expansion estimates from previous studies likely resulted from natural variability and reanalysis data error, indicating the impact of data quality on Hadley cell extent estimates (Staten et al., 2020). Regional variation in tropical expansion has been found to vary substantially by season (Grise et al., 2018) possibly due to natural variability or forcings other than $CO_2$ (Staten et al., 2019).

Many methods have been created to determine the latitude of the poleward edge of the HC (Chen et al.,

2002; Fu et al., 2006; Hudson et al., 2006; Hu and Fu, 2007; Lu et al., 2007; Seidel and Randel, 2007; Previdi and Liepert, 2007; Seidel et al., 2008; Hu et al., 2011; Staten et al., 2011; Zhou et al., 2011; Choi et al., 2014; Karnauskas & Ummenhofer, 2014). Of these methods, the meridional stream function 500 hPa zero-crossing (SF) is the most frequently used due to its direct representation of the zonal-mean HC. Other commonly used metrics such as the sea-level pressure maximum (PSL), the zonal surface wind

zero-crossing (UAS), the subtropical jet maximum (STJ), the eddy driven jet maximum (EDJ), and the precipitation minus evaporation zero-crossing (P-E) are thought to measure the HC extent more indirectly (Waugh et al., 2018). Previous research has studied the trends associated with the many HC extent metrics as well as the correlation and physical links between metrics in reanalyses and climate simulation (Davis and Birner, 2017; Seviour et al., 2018; Waugh et al., 2018). However, to date no study has

analyzed the impact of reanalysis data error on HC width uncertainty. Without consideration of the reliability of the data used for each metric it is not possible to determine the uncertainty in the trends or thoroughly analyze the disagreement between metrics.

HC extent is typically studied using zonal mean data, which is then temporally averaged over either a season or a year, removing the impact of longitudinal variation and short-lived storm systems (Staten et

al., 2019). Reanalysis data is often used to study observed circulation changes as it uses observational data to estimate historical atmospheric conditions, spans multiple decades, and is spatially and temporally continuous. Climate model data is commonly used as well, though some natural variation is not present, and important forcings such as the Pacific decadal oscillation and aerosols are often not accurately modeled (Allen et al., 2014).

In the present study we use data from ERA5, a modern high-quality, high-resolution reanalysis dataset (Hersbach et al., 2020). The ERA5 reanalysis provides continuous data from 1950, though here we will only use data from 1979 onwards to align with previous studies. ERA5 is one of the only reanalysis products which is nearly mass conserving, avoiding the questionable meridional circulation found in other

reanalyses which are not mass-conserving (Davis & Davis, 2018). Since reanalyses combine observations

and modeling to produce estimates of atmospheric variables, observation density and quality impact the

reliability of reanalysis estimates. As a result, the reliability of a particular tropical extent metric depends

partially on the input data error at the relevant location. Moreover, each metric has a specific sensitivity to

the input data error which depends on the meridional gradient in the underlying data near the position

where the metric is defined. Weak gradients around the region of interest result in less robust estimates, as

small errors in the underlying data can result in large uncertainties in the position of the tropical edge.

Unlike most other reanalysis products, ERA5 includes nine ensemble members to allow for a

quantification of uncertainty. Comparing the ensemble members to the standard ERA5 product showed

that these two products produce similar tropical extent time series. The ensemble members use the same

observations and data assimilation scheme, but with slightly less precision (Hersbach et al., 2020). These

ensemble members are produced by introducing slight perturbations in observations and model

parameters within their respective error ranges (Isaksen et al., 2010), meaning that the ensemble spread

does not represent other sources of error such as structural uncertainty, and is therefore only a portion of

the actual uncertainty (Tebaldi and Knutti, 2007). Because reanalysis products use different observations

and model configurations, the inter-reanalysis spread which previous studies have analyzed (Davis and

Rosenlof, 2012) is much larger than the ERA5 ensemble spread and may overstate the uncertainty,

particularly when older reanalyses are included. While the ERA5 ensemble underestimates the

uncertainty, it does allow for a systematic analysis of the uncertainty in a single reanalysis product which

may represent the relative uncertainties in each metric more accurately.

To date no study has quantified the impact of the data uncertainty in a single reanalysis product on HC

extent trend estimates, resulting in a lack of information about this source of error in HC extent metrics.

Due to this knowledge gap, previous studies have not been able to consider the reanalysis data error

differences between metrics, or the sensitivity to this error. In the present study, we provide this

information by analyzing the ERA5 ensemble members, as this allows for a systematic analysis of data

uncertainty and its impact on HC estimates. In utilizing these ensemble members, we provide estimates of

the uncertainty in the reanalysis data used by each metric and the sensitivity of each metric to these data

errors. This study further aims to provide a range of observed HC extent trends for the various metrics in

each hemisphere and season using the modern ERA5 reanalysis dataset.

The structure of the paper is as follows. In Section 2 we describe the methodology, including the HC

extent metrics to be analyzed. In Section 3, we start by measuring the HC trend uncertainty of all metrics

in each season. Next, we determine the annual variation in HC extent for all six metrics in each season.

Subsequently, we attempt to determine the causes of uncertainty in the SF estimates. Following these

results, in Section 4 we provide suggestions for determining the correct HC extent metrics to use in future

studies.

**2   Data and methods**

### 2.1   Data

ERA5 monthly averaged ensemble member data from March 1979 through February 2022 were acquired

for use in this study. March is selected as the starting month as meteorological winter spans multiple

calendar years, meaning that March 1979 is the start of the first complete season, while February 2022 is

chosen as the final month to allow for the maximum number of complete seasons. ERA5 contains 9

ensemble members and one control member. Because the control member is produced through a more

thorough data assimilation process, involving slightly more precision, the control was excluded from the

ensemble analysis. The ensemble members are created by perturbing the observations and model

tendencies. As these ensemble members only represent a portion of the actual uncertainty, compared to

the interannual variation and inter-reanalysis variation these differences are small. Because the ensemble

spread in ERA5 is smaller than the actual uncertainty, we will primarily focus on the relative uncertainties

between the different metrics. Acquired pressure level data includes the meridional and zonal wind fields,

while surface level data included 10-m zonal wind, sea-level-pressure, precipitation, and evaporation. This data is used to compute the tropical extent for the six metrics described in the next sub-section. Data from these 9 ensemble members are regridded using a first-order conservative regridding from a 0.5-degree resolution to a 1-degree resolution using the Climate Data Operator (CDO) 'remapcon' function (Schulzweida, 2022). The 1-degree and 0.5-degree resolution data produced similar tropical extents. All ensemble members are then zonally and seasonally averaged into March-April-May (MAM), June-July-August (JJA), September-October-November (SON), and December-January-February (DJF). Separately annual means are calculated for each year from March-February.

## 2.2 Methods

Six different tropical width estimation methods are used, each covering different atmospheric regions near the HC edge or using different atmospheric variables. Each method is computed using the Python version of TropD, a module which estimates HC width from zonally averaged atmospheric data, yielding a latitude estimate for the poleward extent of the tropics in each hemisphere (Adam et al., 2018). PyTropD uses spline interpolation to produce more latitudinally precise estimates of tropical extent than the 1-degree resolution. All tropical width metrics use the standard configuration of TropD.

### 2.2.1 Meridional Stream Function

The meridional stream function 500-hPa zero-crossing (SF) is chosen as it is the most commonly used metric. This method uses the meridional wind to estimate the edge of the zonal-mean meridional circulation. The stream function values are calculated by integrating the zonal-mean meridional wind at each latitude from the top of the atmosphere to the level of interest. From these stream function values, the NH (SH) 500 hPa zero-crossing North (South) of the minimum (30 degrees) and South (North) of the maximum (60 degrees) is determined as the poleward edge of the tropics in each hemisphere.

### 2.2.2 Subtropical Jet

The subtropical jet adjusted peak between 100 and 400 hPa (STJ) is presented here as it requires upper-troposphere zonal wind, which directly contrasts with the upper-tropospheric meridional wind used by the SF method. Substantial issues exist with the STJ metric, and it has been shown to poorly correlate with other tropical extent metrics (Waugh et al., 2018).

### 2.2.3 Eddy Driven Jet

The eddy driven jet maximum at 850 hPa (EDJ) uses lower-troposphere zonal wind data and is located substantially poleward of the other metrics, but has been shown to be closely linked to other metrics of tropical expansion (Solomon et al., 2016; Davis and Birner, 2017).

### 2.2.4 Precipitation Minus Evaporation

The precipitation minus evaporation zero-crossing (P-E) utilizes surface-level precipitation and evaporation, which are both known to be poorly estimated by reanalyses. Furthermore, the P-E metric is not well connected to other expansion metrics (Seviour et al., 2018). Regardless, this metric is still occasionally used in the literature as it attempts to directly measure the most impactful societal effects of tropical widening.

### 2.2.5 Surface Zonal Wind Zero-Crossing

The surface zonal wind zero-crossing (UAS) uses 10-m zonal wind data and has been shown to be well-correlated with and dynamically linked to the SF metric (Davis and Birner, 2017).

### 2.2.6 Sea-Level-Pressure Maximum

The subtropical sea-level-pressure maximum (PSL) reflects the surface pressure anomaly generated at the descending branch of the HC. This metric was found to be well correlated to SF, moderately correlated to P-E, and very highly correlated to UAS (Waugh et al., 2018).

### 2.2.7 Additional Methods

Notably absent from these metrics are the tropopause break height, which is avoided because it is closely related to the subtropical jet (Davis and Birner, 2017), and outgoing longwave radiation-based metrics, which are avoided due to infrequent use in recent studies. As a result of these decisions, SF, STJ, EDJ, P-E, UAS, and PSL will be analyzed in subsequent sections, with a particular emphasis on SF as it is the most frequently used metric.

To quantify differences between ensemble members multiple statistical methods are used. The ensemble standard deviation (STD) is calculated by taking the STD of the nine ERA5 ensemble members for a given quantity. In Section 3.3, a normalized ensemble STD is calculated by normalizing this quantity by the interannual STD of the ensemble average, and then multiplying by 100 to present the ratio as a percent. This normalized ensemble STD reflects the ensemble spread as a percentage of its interannual variability and is unitless, allowing for comparison between variables.

Kernel density estimates are used to estimate the probability of different Hadley cell expansion rates for each metric. This method, which is described in detail in Silverman (2018), uses kernels to produce smooth nonparametric density estimates. The kernel density estimates are implemented in Python using the Gaussian kernel density estimate function in SciPy (Virtanen et al., 2020) with the standard configuration. The standard deviation of the smoothing kernel is determined from the number of data points and dimensions following Scott's Rule (Scott, 1992).

Because of the limited number of ensemble members present in ERA5 there is substantial uncertainty in ensemble quantities such as the STD. Following the equation for the fractional uncertainty in the STD: $\frac{\partial \sigma}{\sigma} = \frac{1}{\sqrt{2(N-1)}}$ where σ is the STD and N is the number of ensemble members, the relative uncertainty in the ensemble STD is 0.25 (Harding et al., 2014).

190

## 3    Results

In the first two results subsections, we analyze uncertainties in HC trends among the six metrics with the

goal of determining the reliability of each metric and the range of trend estimates. In the subsequent

sections, we examine the SF metric more in depth as it is the most commonly used metric in the literature.

### 3.1    HC trend uncertainty

To quantify the HC trend uncertainty, trends are calculated for each ensemble member using all six

metrics over the period from 1979-2022. From these trends, kernel density estimates are computed in

each season and the annual mean, for all six metrics in the Northern Hemisphere (NH) (Fig. 1) and

Southern Hemisphere (SH) (Fig. 2). A brief description of kernel density estimates is presented in Section

2.2.7.

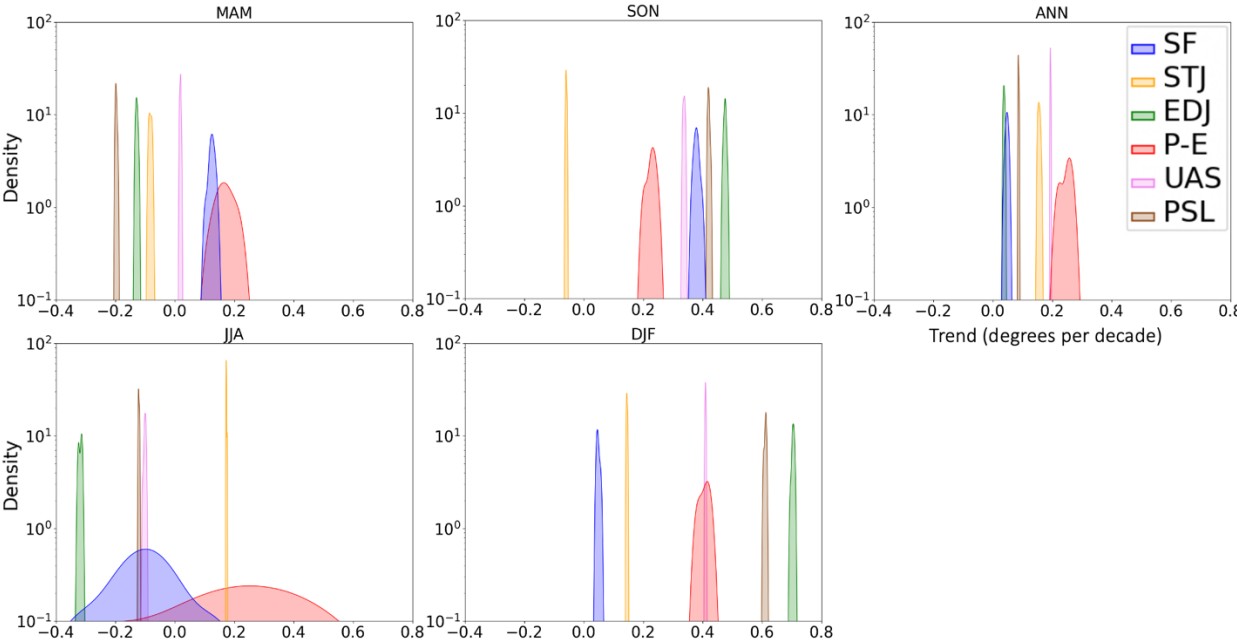

**Figure 1** Tropical widening trends over the NH. Shown are kernel density estimate (y-axis) of tropical

extent trend (1979-2022) in degrees per decade (x-axis), computed from the nine ERA5 ensemble

members for all seasons and the annual mean. Density indicates the relative proportion of trends equal to

a particular value, with the area under the curve equal to 1. Note the logarithmic scale for density on the

y-axis due to the large differences between metrics. Positive x-axis values represent northward trend in tropical extent.

While in individual seasons the trends vary substantially between metrics, the annual mean trends are typically more similar. The NH annual mean trend for SF is roughly 0.05 degrees per decade, while other metrics estimate 0.05 to 0.3 degrees per decade. The SH annual mean trend ranges between 0.1 and 0.3 degrees per decade, with most metrics (including SF) estimating 0.2 degrees per decade. The near-zero NH tropical expansion rate and weak SH expansion are similar to the generally weak expansion found across multiple reanalysis datasets in Grise et al. (2019). It is also of note that the rate of expansion for the annual mean of P-E in the NH is larger than all other metrics.

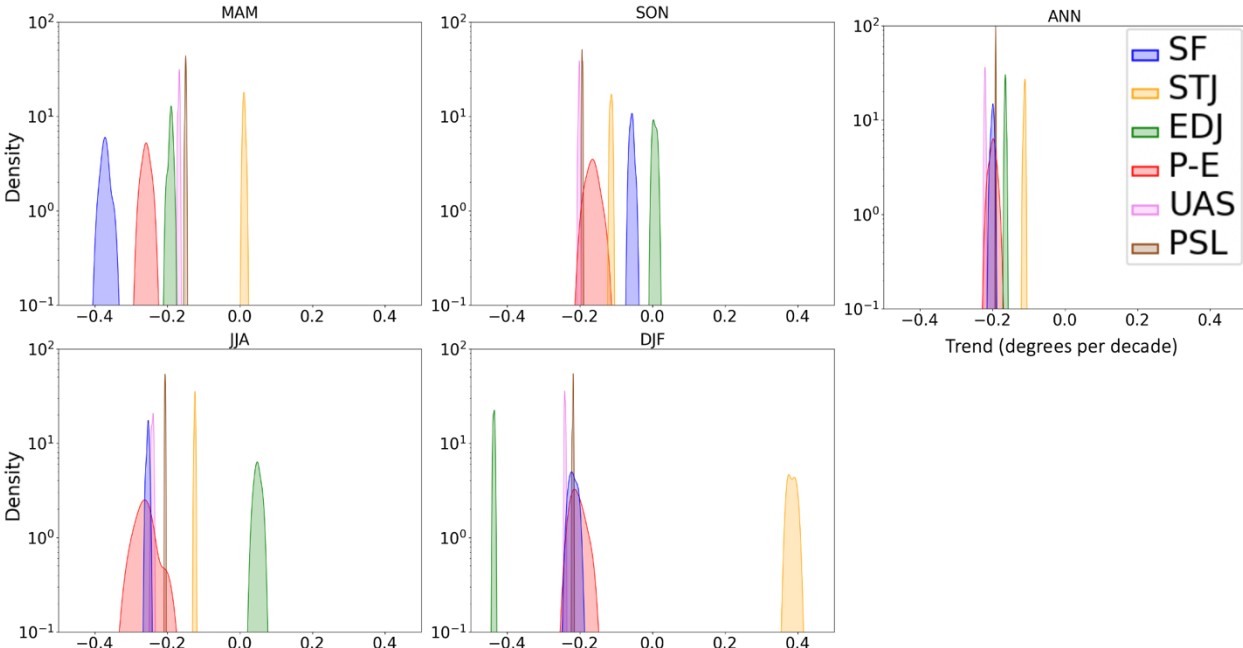

**Figure 2** Same as Fig. 1 for SH. Note the different x-axis values, with negative values representing southward expansion.

On seasonal time scales, the trends and their uncertainty vary substantially between the different metrics, as shown by the kernel density estimates and the ensemble trend STD; a table of the latter is presented in the Supplement (Table S1). P-E features the largest uncertainty, particularly In the NH and in JJA where the uncertainty is roughly 2 orders of magnitude greater than other metrics. Over the NH, the SF exhibits the second least certain trend in all seasons, particularly in JJA, where the range of estimates include both

poleward and equatorward trends and is nearly one order of magnitude less certain than other metrics. Considering that the ensemble spread underestimates the actual uncertainty, the presence of both positive and negative trends indicates that in JJA in the NH SF is not a reliable metric. Over the SH, SF estimated tropical expansion is more robust, as shown by the consistency of the negative sign in all seasons as well as the smaller uncertainty. While the uncertainty over the SH is small for STJ and EDJ in most seasons,

STJ in DJF and EDJ in JJA are substantial outliers in trend, estimating tropical contraction while all other metrics record expansion.

We hypothesize that data uncertainty is a major contributor to the larger uncertainties in SF and P-E. SF is based on poorly observed and poorly constrained meridional wind, and P-E uses imperfectly modeled evaporation and precipitation data. On the other hand, STJ and EDJ are based on better observed and,

constrained (through the thermal wind relationship) zonal wind data, and UAS and PSL are also well-observed and constrained quantities.

### 3.2 HC extent uncertainty

We next examine the HC extent uncertainty in individual years. In doing so, the impact of improved

observations over time can be seen along with the potential influences of internal climate variability on tropical width estimates. The yearly ensemble STD of tropical extent is calculated for each metric in the NH and SH (Fig. 3). P-E has the greatest STD in nearly all seasons and is particularly unreliable in the NH JJA, where the uncertainty is roughly 2 orders of magnitude greater than the four reliable metrics. SF

uncertainty is typically one order of magnitude larger than the four most reliable metrics and is

particularly unreliable in the NH JJA. Over the SH, the SF extent uncertainty is largest relative to the

other metrics. We also note that there is not much interannual variation in STD for the six metrics in

either hemisphere, indicating that internal climate variability does not have a substantial impact on

uncertainty. Counter to this observation are SF and P-E in the NH in JJA, which are even undefined in

some years, resulting from P-E not recording a zero-crossing near the poleward edge of the tropics, and

the SF not recording a closed circulation cell. Examples of the absence of a closed HC in 2019 and 2020

are included in the Supplement (Fig. S1). As a result, in these years the STD is calculated excluding

undefined ensemble member values, resulting in imperfect estimates of variation. It is interesting that

2019 and 2020 are the only two years with some members not reproducing closed NH summer circulation

cells, and it remains to be seen whether this is part of a systematic longer-term trend.

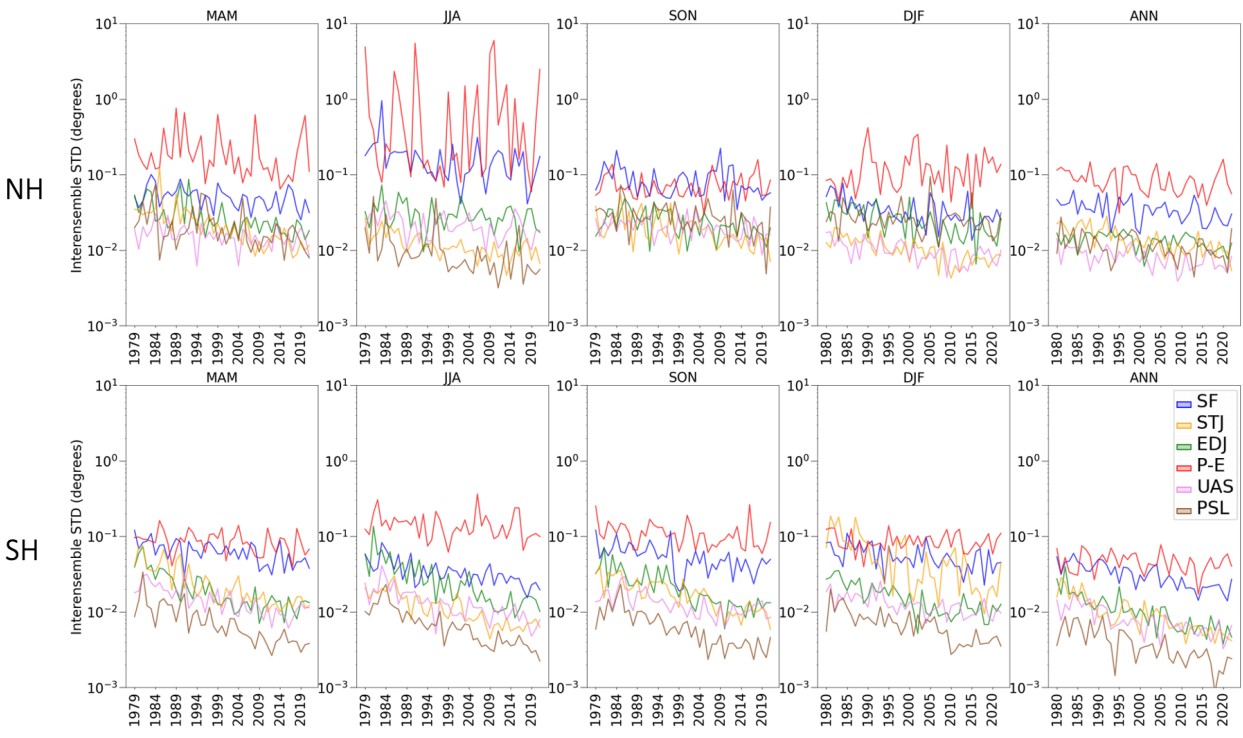


**Figure 3** Interannual variation in tropical extent uncertainty. Shown are annual ensemble STD of HC

extent (degrees latitude) for NH (top) and SH (bottom) by season for all six metrics. Note the logarithmic

scale for the y-axis. The start year for each season is the first available year.

Most notably, Fig. 3 shows a reduction in uncertainty over the 40-year period in most seasons and for

nearly every metric, likely due to the increased quantity and quality of observations assimilated into the

reanalysis. Because the ensemble members are created by perturbing the model parameters along with

observations, both a lack of observations and the presence of lower quality observations increase the data

uncertainty. The metrics based on u-wind and sea-level pressure (STJ, EDJ, UAS, PSL) have consistently

small uncertainty in all seasons, years, and hemispheres, though the variation for PSL especially in the SH

is the smallest of all metrics. Despite the presence of more conventional observations in the NH,

particularly in the earlier decades, the extent uncertainty is smallest in the SH for all metrics, indicating

that both observations and regional dynamics are relevant. The greater uncertainty in the NH may be due

to the abundance of ocean in the SH and topographic variability in the NH, both of which impact the

atmospheric processes which are measured by the HC extent metrics.

### 3.3  Spatial variability of SF data uncertainty

Because the SF is the most widely used metric, the following sections will take a closer look at the

magnitude and impact of data error for just the SF. To quantify the spatial uncertainty of SF data, the

ensemble STD of SF is calculated for all years using zonally and annually averaged meridional wind data.

These annual values are then averaged over two 20-year periods and normalized as described in the

Methods section. This results in the annual mean SF normalized ensemble STD over two time periods

(Fig. 4). The two time periods are 1979-2000 and 2001-2021, comparing the period with fewer

observations to the better observed period following the deployment of many remote sensors (Hersbach et

al., 2020). The normalized ensemble STD during the earlier period is clearly larger than during the later

period, indicating the role of improved observations in the reduced SF extent uncertainty in recent

decades seen in Fig. 3. As the SF metric used here calculates the tropical extent using the 500 hPa SF

values between the Hadley and Ferrel cells, only the improvements in these regions are impactful, while the large decrease in normalized STD in other regions is not directly relevant. However, the

improvements in observations in these regions, particularly the inner tropics, constrain the meridional circulation, likely contributing to reduced uncertainties in other regions of the Hadley circulation in recent decades. As can be seen from Fig. 4, the normalized STD improves from roughly 6% to 4% at 500 hPa at the tropical edges in both hemispheres between the two time periods, though these changes are not statistically significant at the 95% confidence level due to the 0.25 fractional uncertainty in STD as

described in Section 2.2.7. While the ensemble variation is generally less than 10% of the interannual variation, the meridional wind uncertainty is nearly one order of magnitude larger than in the zonal wind used by STJ and EDJ (Fig. S2). Larger normalized uncertainty in the data underlying the SF metric causes larger error in SF-derived HC expansion rates.

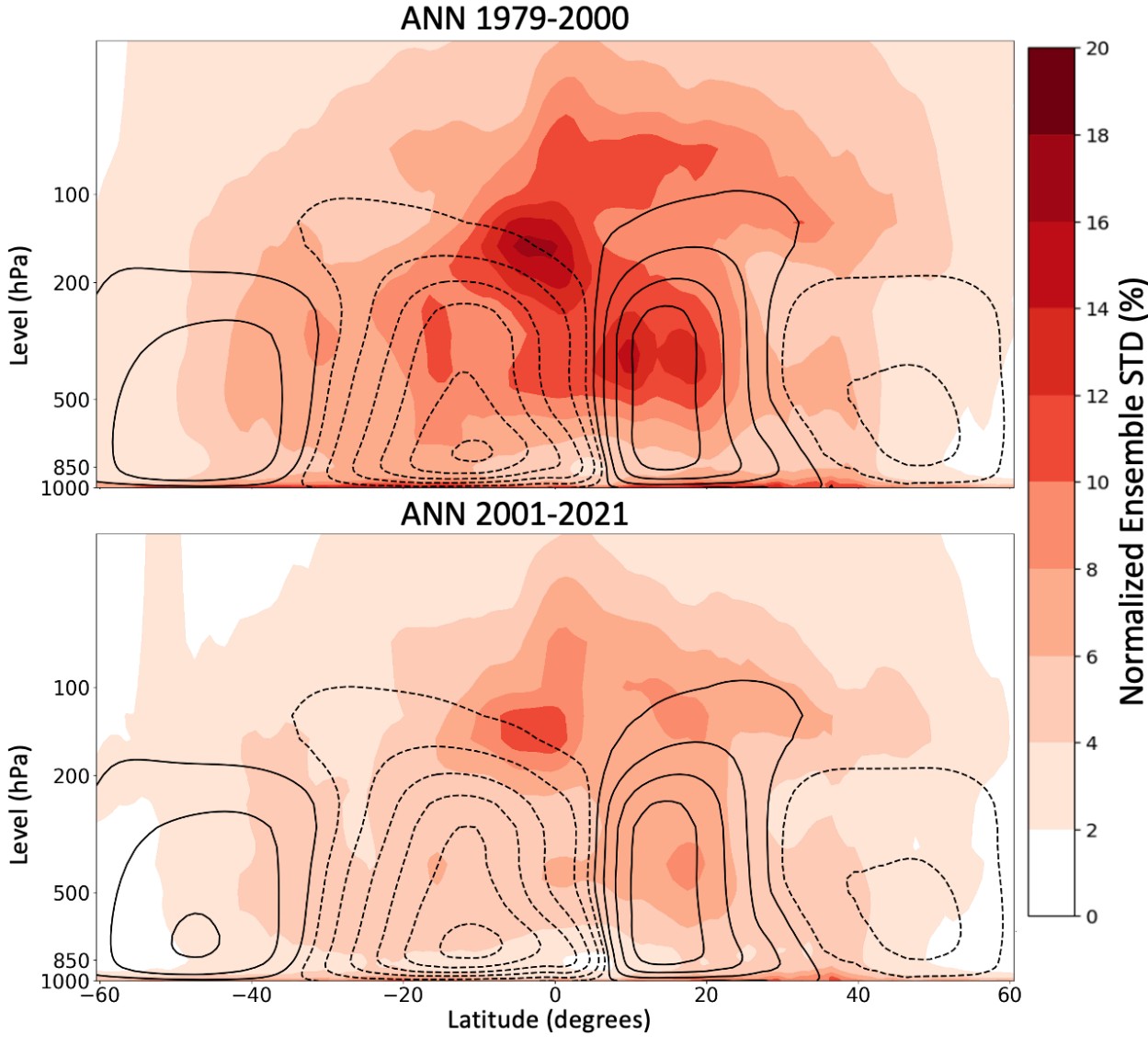


**Figure 4** Meridional cross sections of SF uncertainty for 1979-2000 (top) and 2001-2021 (bottom), demonstrating improvement in SF estimates in recent decades. Shading shows the 20-year average of the annual mean SF normalized ensemble STD (%). Dashed lines indicate negative (counterclockwise) circulation while solid lines indicate positive (clockwise) circulation.


### 3.4 HC extent errors due to weak SF gradients

As shown earlier in this study, in Section 3.2, the SF extent uncertainty is greater in the NH than SH and is particularly pronounced in the NH in JJA. Here, we aim to investigate the impact of the HC structure on SF extent uncertainty. In order to calculate the tropical extent, the latitude where the circulation changes from clockwise to counterclockwise at 500 hPa must be determined. Observational errors will have a larger impact on SF extent estimates if the region of near-neutral circulation between the Hadley and the Ferrel cell is large, as small variations between the ensemble members can cause the zero-crossing to occur over a larger latitude range. We define $\Delta$ in each hemisphere as the width in degrees latitude of the region in the vicinity of the HC edge (poleward of 20 degrees and equatorward of 50 degrees) at 500 hPa where the SF is very weak ($\pm 5*10^9$ kg s$^{-1}$). This $\Delta$ is larger in the NH than SH and is particularly large in NH JJA, coinciding with the seasons and hemispheres where SF trend and extent uncertainty are greatest (Fig. 5).

As shown by Fig. 5, the HC over the NH has a smaller latitudinal extent and features a distinctive narrowing in the middle and upper troposphere. This narrowing coincides with a large $\Delta$, which is not present in other seasons or in the SH. The wide $\Delta$ likely represents the impacts of large land masses and greater longitudinal variation in meridional circulation in the NH (Hoskins et al., 2020). In both hemispheres, however, the region of near-zero circulation in summer is larger than winter, suggesting a possible relationship between the weak summer HC and meridional gradients at the poleward edge of the HC. We speculate that $\Delta$ is impacted by both the zonal variation in circulation and the strength of the meridional circulation. Hence, in JJA in the NH, the weak meridional circulation and large zonal variation in circulation combine to create a persistent large $\Delta$ which causes the SF metric to perform poorly. It is also worth noting that the latitudinal width of the near-zero circulation region does not change significantly during the 40-year period, suggesting that this is a typical and quasi-permanent feature of the atmosphere, and is unrelated to measurement errors.

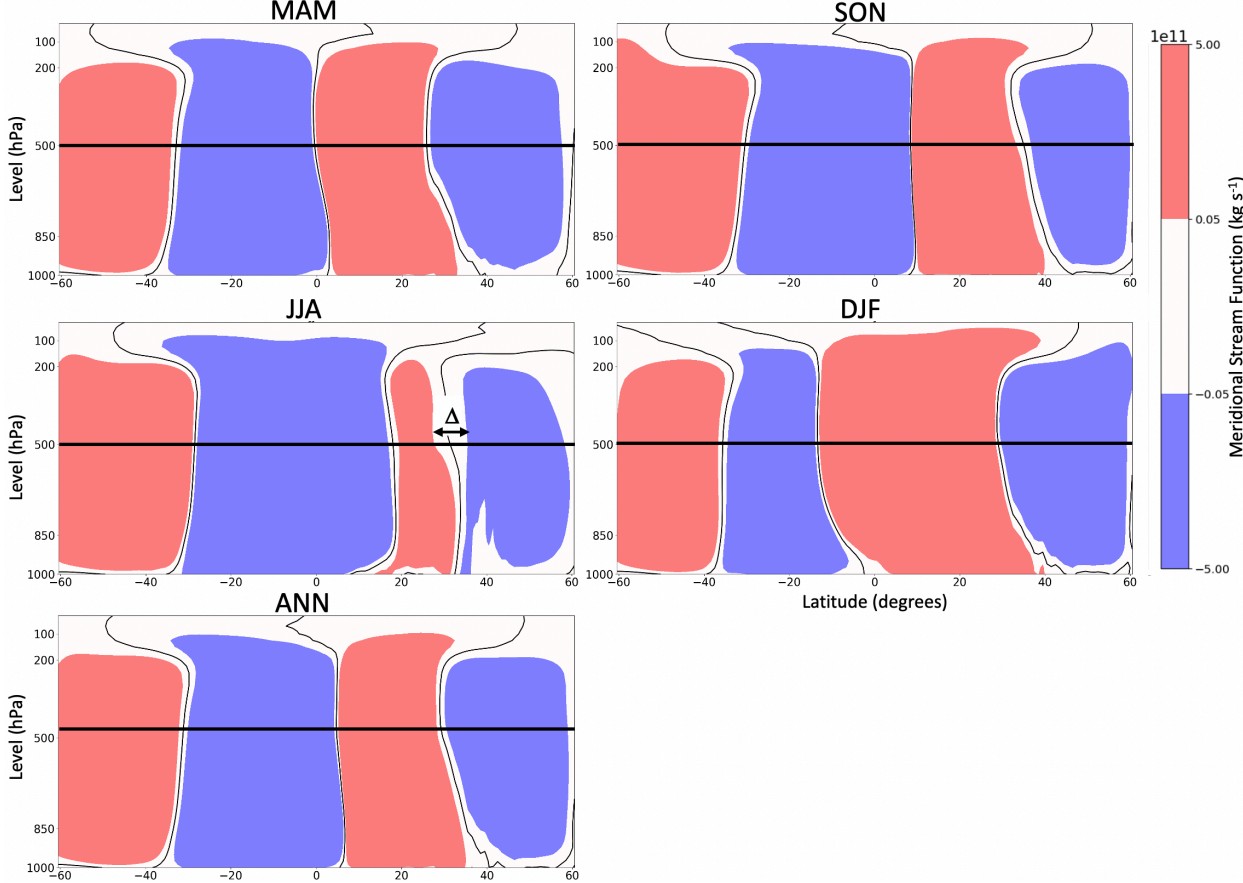

**Figure 5** Climatological mean stream function, highlighting regions with values near zero. Red shading represents clockwise circulation stronger than $5*10^9$ kg/s, blue shading indicates counterclockwise motion stronger than $5*10^9$ kg/s, and white shading the region of near-zero stream function values. Thin black contours denote a zero SF value, while the thick horizontal black line indicates the 500 hPa level where the tropical extent and $\Delta$ are calculated. In JJA an illustration of $\Delta$ is shown in the NH near 500 hPa.

## 3.5 Impacts of data error and weak gradients on SF uncertainty

In the previous sections, the SF extent uncertainty was shown to be related to both data error and weak meridional gradients. We next determine the impact of these two factors on seasonal and annual HC estimates. The large NH JJA and SON tropical extent uncertainty coincides with large $\Delta$ values, suggesting that $\Delta$ plays a significant role when using the SF to estimate tropical width (Fig. 6a). Here, $\Delta$ is

computed from the 1-degree regridded ERA5 data, resulting in relatively coarse latitudinal estimates.

Although data uncertainty, as shown earlier (Fig. 4), partially explains the poor performance of SF relative to other metrics (Figs. 1 and 2) as well as the improvement over time (Fig. 3), no significant relationship was found between the climatological averages of uncertainties in SF data and SF-derived HC extent; a scatter plot showing the poor correlation is included in the Supplement (Fig. S3).

However, when looking at individual annual means, the uncertainty of the SF-based HC extent is well-

approximated as a linear function of the average of the SF ensemble STD in the vicinity ($\pm 2$ degrees latitude) of the HC edge at 500 hPa (Fig. 6b). In the NH, the extent uncertainty is larger than in the SH, likely due to the aforementioned complicating influences of the NH land masses. The extent uncertainty is well correlated with the SF uncertainty and poorly correlated with $\Delta$, indicating that the reduction in data error is the main reason for the decrease in SF extent uncertainty seen before (Fig. 3); a scatter plot

showing the poor fit between $\Delta$ and the annual mean SF extent uncertainty is included in the Supplement (Fig. S4). The improved observations in recent decades thus result in more precise tropical width estimates relative to earlier decades.

The uncertainty of the SF-derived HC trend has no discernable relationship with the SF data uncertainty when comparing seasons (Fig. S5). However, the trend uncertainty is greatest in seasons and hemispheres

where $\Delta$ is largest, though this is mostly noticeable in NH JJA (Fig. S6).

Following this analysis, we conclude that $\Delta$ represents a persistent factor for the uncertainty in the SF metric, which varies primarily by season and hemisphere, and is nearly constant over the years. As discussed earlier, $\Delta$ can be seen as a sensitivity of the HC extent to the data uncertainty, as a larger $\Delta$ results in a larger range of latitudes where data error could cause SF to be zero. $\Delta$ in individual seasons

and hemispheres does not vary much from year-to-year as it is mostly a consequence of relatively fixed meteorological and geographical factors that control the structure of the HC. As a result, $\Delta$ represents a

persistent amplification of data error, resulting in greater extent uncertainty in certain seasons and hemispheres, in particular JJA in the NH.

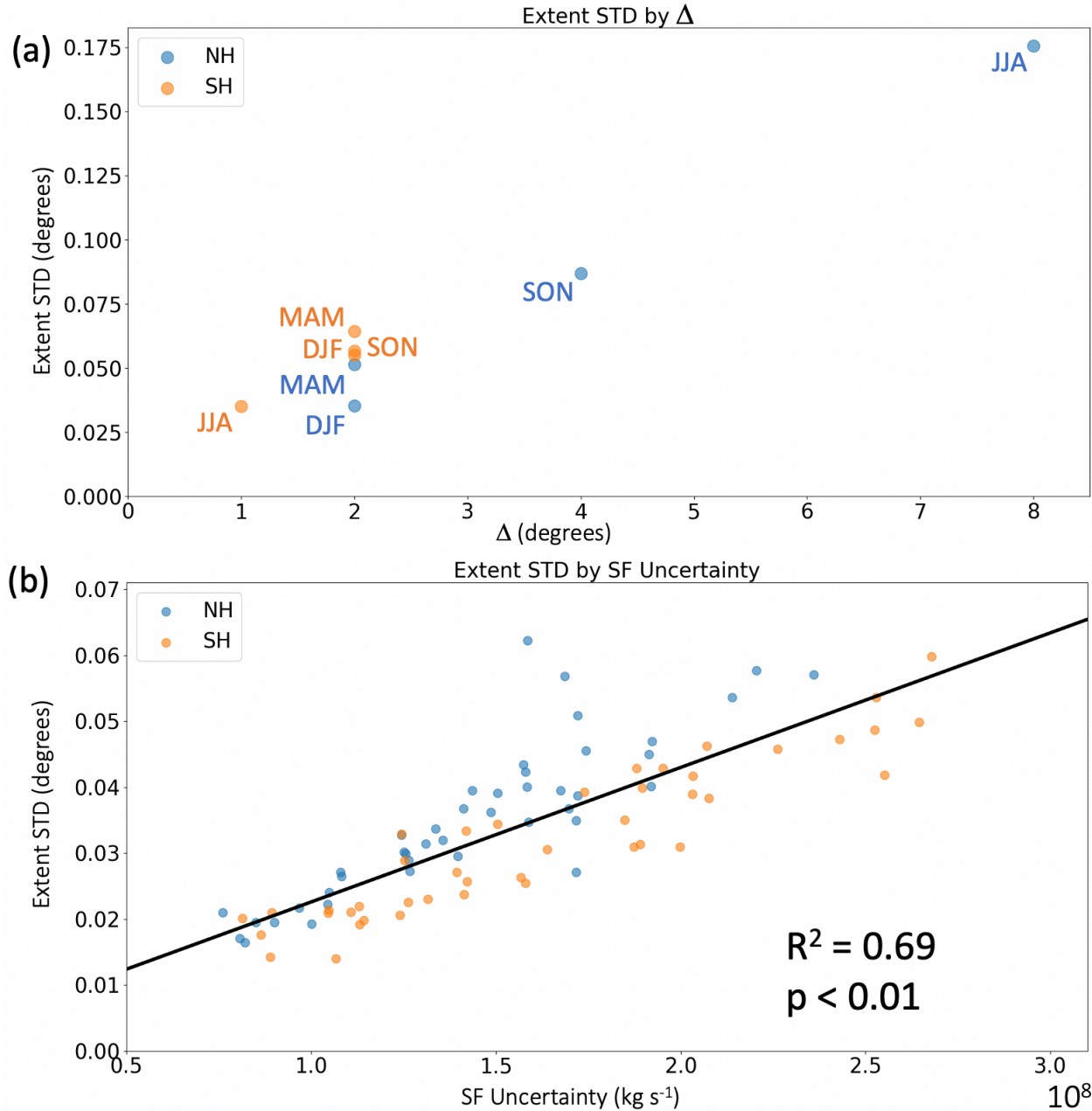

**Figure 6** Uncertainty in the SF-based HC extent. (a) climatological mean of HC extent uncertainty (degrees latitude) by $\Delta$ (degrees) in all four seasons in both hemispheres. $\Delta$ is defined as the width of the near-zero circulation region at 500 hPa in the vicinity of the HC edge, as shown in Fig. 5. (b) Annual

mean HC extent uncertainty (degrees latitude) by the average of the SF ensemble STD at 500 hPa

averaged within 2 degrees of the HC edge (kg s$^{-1}$). A linear best fit line is presented in black.

## 4    Conclusions and discussion

In this study, we used ERA5 ensemble members from 1979-2022 to quantify the uncertainty and long-term trends of different Hadley cell (HC) extent metrics. The annual mean SF HC expansion rate in the

NH was found to be approximately 0.05 degrees per decade in the NH, and 0.2 degrees per decade in the

SH. The HC extent trend uncertainty analysis showed the high uncertainty of SF and P-E trends relative

to the other metrics, particularly in the NH, and most prominently in JJA in the NH. Over the roughly 40-

year reanalysis record, we found substantial improvements in HC extent error for all metrics (Table S2),

especially in the SH, likely due to increases in the number and quality of observations assimilated by the

ERA5 reanalysis system (Figs. 3 and 4). The seasonal and hemispheric differences in SF-derived HC

extent uncertainties were well described by variations in the strength of meridional gradients in SF at the

poleward edge of the HC (Figs. 5 and 6a). Differences in annual mean SF extent uncertainty were well

correlated with the SF uncertainty near the poleward edge of the HC (Fig. 6b). The high sensitivity to data

uncertainty combined with substantial SF data error suggests that the SF metric is not the most reliable

method of determining the width of the HC in reanalyses.

In the following paragraphs, we discuss our findings to provide suggestions for future studies. For each

metric, the data-driven trend and extent uncertainty, agreement with other metrics, and findings from

previous studies will be considered.

Despite being the most widely used metric of tropical extent, SF was found to have much higher

uncertainty in both trend and extent relative to STJ, EDJ, UAS and PSL. The meridional wind used in the

SF metric is generally weak and likely less well-observed than the zonal wind used in other metrics.

Zonal-mean meridional flow is also not dynamically constrained by the temperature field through the

thermal wind relationship as is the case for the zonal wind field. As a result, meridional wind is less accurately represented in reanalysis than zonal wind or surface variables, leading to larger uncertainties relative to the variables used by other metrics. This issue is compounded by the presence of wide regions of near-zero meridional overturning, resulting in a greater sensitivity of the HC edge latitude to the already large data uncertainty. As a result of these two issues, and despite the near-ubiquitous usage in the literature, the SF is not the most useful metric when analyzing tropical extent trends in reanalyses, particularly if the time-period of interest is prior to 2000 or the focus is the NH summer. Over recent decades, when analyzing annual averages, SF becomes a more reliable metric, particularly in the SH.

P-E has the greatest variation between ensemble members in both HC trend and extent, particularly in the NH during JJA. P-E has high uncertainty mostly due to the well-documented issues in modeling precipitation and evaporation in reanalyses (e.g., Simmons et al., 2010). This metric is further challenged by somewhat poor correlations between meridional circulation and vertical moisture flux at the descending branch of the HC, particularly over land masses (Schmidt and Grise, 2017). Despite P-E and related metrics such as the precipitation minimum providing the most societally impactful information, the uncertainty in these metrics is too high to be useful for HC trend analysis with reanalysis data, particularly when other more reliable metrics exist which can be used as proxies.

While STJ featured very small uncertainty, it is not well correlated with SF or P-E, the most direct and relevant metrics of tropical extent, which causes issues when analyzing tropical expansion (Davis and Birner, 2017).

PSL was shown in Davis and Birner (2017) to be well correlated with SF and moderately correlated with P-E, suggesting that it functions as a useful proxy for the societally impactful effects of HC extent change. In our study the uncertainty for PSL tropical extent was found to be very small both in trend and extent. In light of these results, we conclude that PSL is reliable and potentially useful.

EDJ featured little uncertainty in most seasons and hemispheres in both trend and extent but resulted in trends outside of the range of the other metrics over most seasons. This indicates that the EDJ, though reliable and well correlated with SF, is also impacted by other processes that are not closely related to the width of the HC, particularly over individual seasons. As a result, EDJ is most useful for analyzing the
mid-latitudes where changes to the EDJ are most relevant and for studying annual mean HC change.

For UAS, both the trend and extent uncertainties were found to be small in all seasons and hemispheres, and the trends were within the range estimated by other metrics. UAS has also been shown to be well correlated to SF and P-E (Davis and Birner, 2017). Because of these factors, we find UAS to be a reliable and useful metric for analyzing tropical extent trends in reanalyses.

This study focused on errors within a single reanalysis dataset, while many previous studies were based on data from multiple reanalyses, climate models, or observations. When analyzing multiple reanalyses, uncertainty arising from measurement error is likely to exist, as many of the issues highlighted in this study are present in all reanalyses. However, additional uncertainties may be present in the inter-reanalysis spread due to variation in observations and assimilation schemes. Our results from the ERA5
ensemble may translate poorly to inter-reanalysis ensembles containing older reanalyses, leading to some metrics which were found to have low uncertainty in ERA5 being less reliable across reanalyses. Some of the issues seen in this analysis, such as the weak meridional gradients in SF are likely to be present in climate model data as well, while others such as observational uncertainty are instead replaced by modeling error. Future work expanding on this topic should quantify the tropical extent uncertainty in
climate models, and in addition determine the most reliable metrics.

**Code availability**

The code for analysis is available at https://zenodo.org/record/7430530.

**Data availability**

This study uses monthly averaged ensemble members from the ERA5 reanalysis, which can be

downloaded at https://cds.climate.copernicus.eu (Hersbach et al., 2020)

**Supplement**

**Author Contributions**

DB and TR designed the study. DB performed the analyses and wrote the manuscript with feedback from

all authors.

**Competing Interests**

The authors declare that they have no conflict of interest.

**Disclaimer**


**Acknowledgements**

We acknowledge the European Center for Medium-Range Weather Forecasts (ECMWF) for producing

the ERA5 reanalysis dataset. We also thank Hans Hersbach and Paul Berrisford at ECMWF for their

communication on the ERA5 reanalysis dataset, and the Center for High Performance Computing at the

University of Utah for providing computing resources.

**Financial Support**

Thomas Reichler acknowledges funding from the National Science Foundation under award no. 2103013.

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
