# Peer review of "Large Uncertainty in Observed Estimates of Tropical Width"

_EGUsphere, 2022_

## Referee Comment (RC1)

Review of

**Large Uncertainty in Observed Meridional Stream Function Tropical Expansion**

Baldassare et al.

**General**

The authors analyze the uncertainty in metrics for estimating the extent of the Hadley circulation (HC) based on nine ensemble members of the ERA5 reanalysis, focusing in particular on the commonly-used streamfunction (SF) metric. The key findings are a reduction over time of uncertainty, which the authors associate with better quality of the assimilated data, and that the SF metric has a relatively high uncertainty due to less observationally constrained upper wind and relatively weak meridional gradients near the zero-crossing latitude, which increase the uncertainty of the SF metric compared to other zero-crossing metrics.

I think the analysis may merit publication. However, I find some critical issues with the methodology and conclusions. In particular, the authors should do a better job of constraining the uncertainty in their results and better contextualize the results. I also don't think that the authors should provide recommendations or try to rank the different metrics, but rather focus on providing the objective uncertainty estimates and discuss the associated implications. Detailed comments are provided below.

**Comments**

1. I partly disagree with the recommendation by the authors to use surface winds instead of the SF metric. Surface winds are affected by many processes and therefore the extent index based on surface winds captures information which may be fundamentally different from SF-based indices. In addition, as shown in the TropD paper (Adam et al.2018), the variance across models in the PSI and UAS metrics is roughly the same. The recommendation by the authors is strictly based on the apparent reduced uncertainty in ERA5. This point should be clarified, and the relevance of their recommendation better outlined. Similarly, there is great variance across reanalyses, which suggests that the uncertainty estimates by the authors are specific to the ERA5 dataset. This should be discussed.

2. Standard deviation (STD) is calculated across 9 ensemble members. The uncertainty in STD is inversely proportional to the number of degrees of freedom ($N$). Specifically, the fractional uncertainty in STD is $\dfrac{\delta\sigma}{\sigma} = \dfrac{1}{\sqrt{2(N-1)}}$ which for $N = 9$ gives an uncertainty of 25% in STD estimates. There is therefore significant uncertainty in the uncertainty estimates based on STD using only 9 members. This is a critical point in the discussion of changes in uncertainty over time. For example, in line 250, the change in STD from 6% to 4% is not statistically significant at 95% confidence.

3. Please better explain the y-axis in figures 1 and 2. It is not clear what these value signify and why the distributions are so smooth.

4. In figures 1 and 2 the trend itself is subject to uncertainties due to natural variability. In this respect, the mean extent is more revealing. Thinking about the trend as proportional to the difference between two mean values, the metric-uncertainty in a trend is therefore simply $\sqrt{2}$ times the uncertainty of the mean in a given period. Basing the estimates on regression trends adds more noise, since there is additional uncertainty associated with the trend estimation.

**Comments by line**

11     I would refrain from using the term 'tropical expansion' as there are recent indications that the tropics are actually narrowing (while 'tropical expansion' is commonly used, it is a bad choice of words since it is the subtropics that are in effect expanding). Hadley cell expansion is the subject of this analysis, and is a more appropriate term in this case.

21     State the period of the calculated trend.

41     in some cases there are conflicting results, but not as a rule.

54-58 This is not correct. Davis and Rosenlof (2012) do an excellent job of demonstrating the variance across datasets and methods. The failure to cite Davis and Rosenlof (2012) is particularly upsetting, as this paper was pivotal in convincing the community that there is a need to better constrain estimates of HC expansion. Similarly, in the TropD paper, variance across models and sensitivity to grid spacing are examined. The authors examine variability within a particular dataset, and should better delineate their analysis from previous works.

65     I don't agree. Chemke and Polvani study HC intensity discrepancies and actually specifically state that reanalysis and model extent trends generally agree.

85-92 Over the past few decades, variance across reanalyses in HC extent estimates has increased significantly (Adam et al. 2014), despite "better data". There is therefore every reason to assume that metric uncertainties vary across reanalyses. In other words, estimates based on the ERA5 ensemble cannot be assumed to generally hold for other datasets.

230   Doesn't this contradict the preceding assumption that the reduction in uncertainty is related to improved data quality?

232   There are more stationary waves in the NH but there is significant transient variability in both hemispheres, so it is not clear that this is a valid argument.

**References**

Adam, O., Schneider, T., & Harnik, N., 2014: Role of Changes in Mean Temperatures versus Temperature Gradients in the Recent Widening of the Hadley Circulation, *Journal of Climate*, **27**(19), 7450-7461

Davis, S. M. and Rosenlof, K. H., 2012: A Multidiagnostic Inter-comparison of Tropical-Width Time Series Using Reanalyses and Satellite Observations. *J. Climate*, **25**, 1061–1078

---

## Author Comment (AC1)

Supplement of

**Large Uncertainty in Observed Meridional Stream Function Tropical Expansion**

**Daniel Baldassare[1], Thomas Reichler[1], Piret Plink-Björklund[2], and Jacob Slawson[2]**

- [1]Department of Atmospheric Sciences, University of Utah, Salt Lake City, UT 84112, USA
- [2]Department of Geology and Geological Engineering, Colorado School of Mines, Golden, CO 80401, USA

Correspondence: Daniel Baldassare (daniel.baldassare@utah.edu)

**Table S1: Intermember Trend STD**

**Intermember Trend STD**

|  |  | MAM | JJA | SON | DJF | ANN |
|---|---|---|---|---|---|---|
| **NH** | **SF** | 0.011 | 0.004 | 0.006 | 0.011 | 0.004 |
| | **STJ** | 0.003 | 0.002 | 0.003 | 0.011 | 0.002 |
| | **EDJ** | 0.005 | 0.009 | 0.006 | 0.002 | 0.002 |
| | **P-E** | 0.011 | 0.027 | 0.017 | 0.018 | 0.009 |
| | **UAS** | 0.002 | 0.003 | 0.002 | 0.002 | 0.002 |
| | **PSL** | 0.002 | 0.001 | 0.001 | 0.001 | 0.001 |
| **SH** | **SF** | 0.010 | 0.118 | 0.009 | 0.005 | 0.005 |
| | **STJ** | 0.005 | 0.001 | 0.002 | 0.002 | 0.004 |
| | **EDJ** | 0.004 | 0.005 | 0.004 | 0.004 | 0.003 |
| | **P-E** | 0.030 | 0.263 | 0.015 | 0.017 | 0.019 |
| | **UAS** | 0.002 | 0.003 | 0.003 | 0.002 | 0.001 |
| | **PSL** | 0.003 | 0.002 | 0.003 | 0.004 | 0.001 |

Table S1: HC trend intermember standard deviation (degrees per decade) in each hemisphere, for all seasons and the annual mean, of all six metrics across the nine ensemble members. Results are qualitatively similar to the kernel density estimates (Fig. 1, Fig. 2) but instead provide a single uncertainty value rather than a range of probable values.

25

**Text S1: Meridional and zonal wind errors**

The normalized error in meridional wind was found to be much larger than zonal wind, as shown in Fig. S1. The larger normalized differences between ensemble members in meridional wind compared to zonal wind indicate that the greater uncertainty in SF compared to STJ, EDJ or UAS may be the result of larger relative errors in the underlying data. The non-normalized intermember STD for meridional wind was found to be of similar magnitude to zonal wind (not shown), with the low absolute values and subsequent smaller time variation resulting in larger normalized errors.

[Figure]

Fig. S1: Annual mean meridional wind (left) and zonal wind (right) normalized intermember STD (%) from 1979-2000 (top) and 2001-2021 (bottom). Note the different scales for meridional and zonal wind due to the much larger errors in meridional wind.

**Text S2: Unenclosed stream function northern hemisphere summer 2019 and 2020**

The NH JJA HC in 2019 and 2020 was found to be unenclosed in some ensemble members, with no positive circulation near 500 hPa and 20 degrees North. In 2019 a majority of ensemble members showed these features, while in 2020 only one member did not have a single enclosed cell in the NH. These were the only two years where this behavior occurs. A variety of climatic oscillations were checked in the attempt to find an explanation for why this only occurs in these two years, but no satisfactory explanation was found. Examples of the unenclosed circulation are shown in Fig. S2. The SF metric is not able to calculate a HC extent in this situation, leading to imperfect estimates of SF uncertainty.

[Figure]

Fig. S2: SF values (kg s⁻¹) for ensemble member 4 in JJA 2019, and ensemble member 1 in JJA 2020, showing an unenclosed NH HC. The black contours indicate the zero-crossing, while shading signifies SF intensity. Note the region near 500 hPa and 20 degrees in both years where there is no positive circulation.

60    **Text S3: Poor correlation between climatological mean SF uncertainty and SF extent**

The climatological mean SF uncertainty and climatological mean SF extent were not found to be well correlated in this analysis, leading to the interpretation that climatological mean SF uncertainty is best explained by $\Delta$. The scatter plot of these two variables is shown in Fig. S3. The SF uncertainty near the tropical edge at 500 hPa does not vary noticeably by season or hemisphere, while the extent STD does.

[Figure]

65

Fig. S3: Climatological mean SF extent STD (degrees latitude) by 500 hPa SF uncertainty (kg s$^{-1}$) near the tropical edge, showing no significant relationship.

70

75

**Text S4: Annual mean SF extent STD and Δ**

When analyzing the annual mean SF uncertainty over individual years, Δ is a poor predictor. The relationship between these two variables is shown in Fig. S4. The width of Δ does not change noticeably during the time period of this study, while the extent STD decreases. Further, Δ is most noticeable in the difference between seasons and hemispheres, and is less important in the annual mean.

[Figure]

Fig. S4: Annual mean extent STD (degrees latitude) by Δ (degrees latitude), demonstrating that Δ is not a useful predictor of SF extent STD in the annual mean.

**Text S5: SF trend STD by 500 hPa SF uncertainty**

Similar to the climatological mean SF extent STD, no relationship was found between the SF trend STD and the 500 hPa SF uncertainty. The trend uncertainty differences between seasons are not well-described by the SF error differences, indicating that the particularly poor performance of SF in NH JJA is unrelated to data error.

[Figure]

Fig. S5: SF trend STD (degrees per decade) by 500 hPa SF uncertainty (kg s⁻¹) for all four seasons in both hemispheres.

**Text S6: SF trend STD by Δ**

Aside from NH JJA, which features both large trend STD and large Δ, the relationship between trend STD and Δ among seasons is unclear. However, it is shown here that the large trend STD in NH JJA is caused by the large Δ values during this season as the small perturbations in reanalysis data are able to cause large differences in trend. Here we can see once again the Δ functioning as a sensitivity to data error.

[Figure]

Fig. S6: SF trend STD (degrees per decade) by Δ (degrees) for all four seasons in both hemispheres.

**Table S2: Improvements in HC extent uncertainty**

As shown in Fig. 3, the HC extent uncertainty improves from 1979-2022 for almost all metrics in every season and the annual mean. This coincides with increasing quantity and quality of observations. Here, we quantify the improvement in the annual mean uncertainty by applying a linear best fit to the time series of intermember extent STD for each metric in the NH and SH. This results in larger improvements for the least certain metrics, P-E and SF, though improvements are substantial for all metrics in both hemispheres.

**ANN Extent Intermember STD Change (degrees per decade)**

|     | SF      | STJ     | EDJ     | P-E     | UAS     | PSL     |
|-----|---------|---------|---------|---------|---------|---------|
| **NH** | -0.0042 | -0.0033 | -0.0011 | -0.0065 | -0.0010 | -0.0014 |
| **SH** | -0.0074 | -0.0037 | -0.0041 | -0.0021 | -0.0020 | -0.0012 |

Table S2: Annual mean HC extent intermember STD change (degrees per decade) for all six metrics in both hemispheres.

---

## Author Response (AR1)

**Large Uncertainty in Observed Meridional Stream Function**

**Tropical Expansion**

**Daniel Baldassare, Thomas Reichler, Piret Plink-Björklund, and Jacob Slawson**

We thank the reviewers for their comments, suggestions, and remarks which have helped to improve this manuscript. We have incorporated most of the reviewer comments, resulting in substantial changes to the text. Major changes include improving the description of the ERA5 ensemble spread and how it compares to the inter-reanalysis spread. The text has also been altered to focus on discussing the uncertainties in each metric rather than providing recommendations. Below we list the reviewer comments, with our responses in red.

**Review 1**

**General**

The authors analyze the uncertainty in metrics for estimating the extent of the Hadley circulation (HC) based on nine ensemble members of the ERA5 reanalysis, focusing in particular on the commonly-used streamfunction (SF) metric. The key findings are a reduction over time of uncertainty, which the authors associate with better quality of the assimilated data, and that the SF metric has a relatively high uncertainty due to less observationally constrained upper wind and relatively weak meridional gradients near the zero-crossing latitude, which increase the uncertainty of the SF metric compared to other zero-crossing metrics.

I think the analysis may merit publication. However, I find some critical issues with the methodology and conclusions. In particular, the authors should do a better job of constraining the uncertainty in their results and better contextualize the results. I also don't think that the authors should provide recommendations or try to rank the different metrics, but rather focus on providing the objective uncertainty estimates and discuss the associated implications. Detailed comments are provided below.

Thank you for your comments. Regarding the uncertainty, we have added language describing how the ERA5 ensemble members are created, as well as more information about the meaning of the uncertainty analyzed in this manuscript. While discussing this we describe what the ERA5 uncertainty does and does not represent, which helps to constrain the uncertainty and contextualize the results. This also allows for better comparison with the inter-annual and inter-reanalysis spread. We agree with the suggestion to avoid recommendations, and have adjusted the language particularly in the conclusion to be less prescriptive.

**Comments**

1. I partly disagree with the recommendation by the authors to use surface winds instead of the SF metric. Surface winds are affected by many processes and therefore the extent index based on surface winds captures information which may be fundamentally different from SF-based indices. In addition, as shown in the TropD paper (Adam et al.2018), the variance across models in the PSI and UAS metrics is roughly the same. The recommendation by the authors is strictly based on the apparent reduced uncertainty in ERA5. This point should be clarified, and the relevance of their recommendation better outlined. Similarly, there is great variance across reanalyses, which suggests that the uncertainty estimates by the authors are specific to the ERA5 dataset. This should be discussed.

   Regarding the recommendation to use UAS rather than SF, this has been removed, and replaced with discussion of the findings from this manuscript and other papers. Language has been clarified in the introduction, results, and conclusion describing how the ensemble spread compares with the inter-reanalysis spread. The inter-reanalysis spread is larger due to the addition of changes in assimilation and observations. The ensemble spread does not reflect the structural uncertainty, meaning that it underestimates the uncertainty. However, the variation in assimilated observations between reanalyses causes an overestimation of uncertainty. As a result, we have changed the text to emphasize our focus on the relative uncertainty between metrics. In addition, we focus on the uncertainty within a single reanalysis, under the assumption that these results will be similar in other reanalyses, but no assumption is made that these results will also apply to the inter-reanalysis spread which includes additional uncertainties. As a result, changes have been made to the conclusion to better contextualize the results.

2. Standard deviation (STD) is calculated across 9 ensemble members. The uncertainty in STD is inversely proportional to the number of degrees of freedom ($N$). Specifically, the fractional uncertainty in STD is $\dfrac{\delta\sigma}{\sigma} = \dfrac{1}{\sqrt{2(N-1)}}$ which for $N = 9$ gives an uncertainty of 25% in STD estimates. There is therefore significant uncertainty in the uncertainty estimates based on STD using only 9 members. This is a critical point in the discussion of changes in uncertainty over time. For example, in line 250, the change in STD from 6% to 4% is not statistically significant at 95% confidence.

   Thank you for pointing this out. We added clarification that these changes are not statistically significant.

3. Please better explain the y-axis in figures 1 and 2. It is not clear what these value signify and why the distributions are so smooth.

   We have added more explanation for the KDE based on this comment and others.

4. In figures 1 and 2 the trend itself is subject to uncertainties due to natural variability. In this respect, the mean extent is more revealing. Thinking about the trend as proportional to the difference between two mean values, the metric-uncertainty in a trend is therefore simply 2 times the uncertainty of the mean in a given period.

Basing the estimates on regression trends adds more noise, since there is additional uncertainty associated with the trend estimation.

We agree with these comments and included section 3.2 to focus on the extent uncertainty partly due to this issue. In many ways the extent uncertainty is a better measure of the reanalysis data error driven uncertainty in each metric. However, because HC trend is more frequently studied than HC extent, we include this analysis as well.

**Comments by line**

11    I would refrain from using the term 'tropical expansion' as there are recent indications that the tropics are actually narrowing (while 'tropical expansion' is commonly used, it is a bad choice of words since it is the subtropics that are in effect expanding). Hadley cell expansion is the subject of this analysis, and is a more appropriate term in this case.

Tropical expansion has been changed to Hadley cell expansion to be more accurate.

21.    State the period of the calculated trend.

Done

41    in some cases there are conflicting results, but not as a rule.

54-58  This is not correct. Davis and Rosenlof (2012) do an excellent job of demonstrating the variance across datasets and methods. The failure to cite Davis and Rosenlof (2012) is particularly upsetting, as this paper was pivotal in convincing the community that there is a need to better constrain estimates of HC expansion. Similarly, in the TropD paper, variance across models and sensitivity to grid spacing are examined. The authors examine variability within a particular dataset, and should better delineate their analysis from previous works.

This reference has been added and the goals and context of the manuscript more accurately described.

65    I don't agree. Chemke and Polvani study HC intensity discrepancies and actually specifically state that reanalysis and model extent trends generally agree.

Width has been changed to circulation to fix this issue

85-92  Over the past few decades, variance across reanalyses in HC extent estimates has increased significantly (Adam et al. 2014), despite "better data". There is therefore every reason to assume that metric uncertainties vary across reanalyses. In other words, estimates based on the ERA5 ensemble cannot be assumed to generally hold for other datasets.

This is an interesting point, and something that we address here in the introduction, as well as in the conclusion. Every reanalysis product has a data-driven uncertainty similar to that analyzed for ERA5 in this manuscript, which we assume is reasonably similar. For most other products this uncertainty cannot be analyzed, so we hope that our results will be useful for future studies involving these other products. The inter-reanalysis spread studied by Adam et al., (2014) is different though, as this spread includes differences in assimilation techniques and observations which are not present in a single reanalysis as studied in this manuscript. Because of this and other comments, we have added clarification in the introduction, results, and conclusion describing how the ensemble spread analyzed here

differs from the inter-reanalysis spread. In general we believe that our results are more applicable to other reanalyses than the inter-reanalysis spread because the uncertainty is more similar.

230   Doesn't this contradict the preceding assumption that the reduction in uncertainty is related to improved data quality?

We have changed the text here to explain that both dynamics and observation quality are significant.

232   There are more stationary waves in the NH but there is significant transient variability in both hemispheres, so it is not clear that this is a valid argument.

This is a good point, we have changed the language from 'likely' to 'may' to reflect that many plausible alternative theories exist.

**References**

Adam, O., Schneider, T., & Harnik, N., 2014: Role of Changes in Mean Temperatures versus Temperature Gradients in the Recent Widening of the Hadley Circulation, *Journal of Climate*, **27**(19), 7450-7461

Davis, S. M. and Rosenlof, K. H., 2012: A Multidiagnostic Inter-comparison of TropicalWidth Time Series Using Reanalyses and Satellite Observations. *J. Climate*, **25**, 1061– 1078

**Review 2**

In this manuscript the authors attempt to estimate the observed uncertainty in recent Hadley cell extent trends/changes as seen in ERA5. For this the authors analyze the spread across ERA5 members in different Hadley cell metrics. Lastly, the authors link the uncertainty in the HC extent to the uncertainty in the magnitude of the circulation around its edge. The overall motivation for this research, as stated in the abstract and introduction, is the different expansion rates across different Hadley cell metrics, reported by previous studies. In this manuscript, however, the authors do not address this issue. In fact, the ERA5 uncertainty of each metric, is not as large as the inter-metric spread, reported before, nor as the trend it self of each metric. So I am not sure that this manuscript helps us better constrain the different Hadley cell expansion rates. Moreover, as discussed in previous studies (which the author cite), the lack of correlation between the different metrics suggests that they represent and driven by different processes, and we thus not necessarily expect to observe the same trend in each metric. Only reporting the uncertainty in the Hadley cell trends (which seem to be small, relative to the signal and the inter-metric spread), in my opinion, is not sufficient for publication. The most important conclusion here is the reduction in uncertainty across ERA5 members over the years. But this is a technical result.

Thank you for your comments. Regarding the inter-reanalysis spread and inter-annual variability which has been widely studied in previous papers, we agree that the original manuscript did not adequately describe the differences in the ensemble spread. As a result of these comments and others, substantial text has been added to the introduction, results, and conclusions to better describe how the data studied in this manuscript differ from these other studies. To summarize, the ensemble spread is created only through perturbations in observations and model parameters about their error ranges, while the inter-reanalysis spread also includes differences in model construction and in

which observations are assimilated. Because of this, our uncertainty is much smaller, and likely underestimates the true uncertainty in the ERA5 reanalysis, as the structural uncertainty is not included. However, the inter-reanalysis spread overestimates the uncertainty because of differences in assimilated observations, particularly when older reanalyses are included. As a result, we focus on the relative uncertainties between metrics, and the language has been clarified in each section to better explain this. The relative uncertainty still demonstrates that SF and P-E have up to 100 times more uncertainty than the 4 high-quality metrics, indicating that these metrics are less reliable, particularly in the NH during JJA and SON. While the absolute uncertainty is not particularly useful, and is now rarely discussed in the text, the differences in uncertainties allow for important differences between metrics to be elucidated. Regarding the comment that we have only reported the uncertainty in Hadley cell trends, we do not believe this is an accurate description of this manuscript. We include the first presentation of the HC trends in ERA5 across multiple metrics. We describe the relationship between reanalysis data uncertainty and metric uncertainty within a single reanalysis for the first time. We also describe the impacts of a region of poorly defined circulation in the NH. Lastly, we show for the first time two seasons with an unenclosed HC.

Below I list several more comments (major and minor):

1. The introduction, in my opinion, should be broaden to give a larger context for this problem, and why its important to investigate the expansion of the circulation. The authors should discuss how/where the Hadley cell is projected to change in coming decades, the mechanisms underlying recent and future Hadley cell changes, and the impacts of such expansion.
Based on this comment we have added more explanation of observed HC expansion, and better contextualized the problem.

2. How much the ERA5 spread is different than large-ensemble spread, which has been documented in previous work (e.g., Grise et al 2019).
We have added substantial text to the introduction describing how the ERA5 spread differs from the inter-reanalysis spread. As the focus of this paper is on reanalysis data we do not compare our results to model data, but do discuss this topic in the conclusion.

3. The missing December in the beginning of ERA5 should not be a motivation to define the annual mean from March to February. This does not allow a proper comparisons to previous work. I suggest using, only for the first year, January and February, for NH winter, and DJF for other years. And use January to December as the canonical definition for the annual mean.
The DJF and JF Hadley cells are quite different, as the Hadley cell in December is similar to the fall HC. Because of this, we do not believe that comparing the DJF Hadley cell to the JF Hadley cell is appropriate. While the offset years are not ideal, we believe that the 2-month offset years are more similar to the standard year than the JF Hadley cell is to the DJF Hadley cell.

4. Please itemize the different paragraphs in the methods section discussing each metric.
Done

5. The Hadley cell extent is usually found by doing an interpolation of the data to a finer grid; have you done the same here?
This has been clarified to explain that spline interpolation is used.

6. In the normalized STD you divide the inter-member spread with interannual variability, but these two may represent different processes. I am thus worried that this metric does not represent a normalized uncertainty.

Yes, it is correct that these two processes are different. We normalize using the interannual variability to analyze the ensemble spread in the context of the interannual spread. Qualitatively, this allows for the ensemble spread to be viewed relative to natural variability, and most importantly allows for the u-wind and v-wind uncertainty to be compared despite wind speeds which are one order of magnitude apart.

7. In Sec. 3.5 the authors argue that the uncertainty in Hadley cell expansion is linked to the gradient of the streamfunction at the Hadley cell edge. However, such link is based on correlation of only eight points, where 5 of them do not follow the regression line, and show no sign of correlation. I am thus not convinced by the authors' arguments, and suggest to remove this analysis along with its discussion.

This is a good point, and this section has been modified as a result. The purpose of this analysis is to point out that the NH JJA and SON Hadley cells have weak gradients, and are thus more susceptible to data error. This section has been rewritten to better explain that the correlation is due to these seasons, and that the weak streamfunction causes issues for these seasons in particular.

**Review 3**

This paper analyzes various "tropical width" metrics from the ERA5 reanalysis ensemble members to assess the "data-driven" uncertainties in the various metrics. As a wide range of widening estimates have been published over the past decade and a half or so, it is very helpful for the community to have a better understanding of the uncertainties in these estimates. The authors use of the ERA5 ensemble members is to be commended, as this available data has heretofore been highly under-utilized. It is encouraging to see the demonstration that, in some cases, the increased observational system in recent decades has led to a smaller spread of tropical width estimates. However, this paper contains numerous over-interpretations of the results presented and fails to compare the spread due to data uncertainty with the much larger spread due to interannual variability. When placed in the proper context, the results of this paper represent an interesting finding regarding relative uncertainty in tropical width metrics and changes over time, but the results are highly oversold as currently written.

Thank you for your comments. From this comment and others we have removed the over-interpretations and focused more on the results rather than interpretation. We do compare the data uncertainty to interannual variability using the normalized ensemble spread, and have also added language to the introduction, methods, results, and conclusion better contextualizing our results.

Most critically, the authors have not made the very obvious comparison between their "data-driven uncertainty" and the uncertainty one gets in the tropical width trends due simply to interannual variability. The authors do use a "normalized intermember STD" in Sect 3.3 (e.g., Fig. 4), and the limited info available from this suggests that data-related uncertainty is actually not that large relative to interannual variability. E.g., The normalized STD near the HC edges at 500 hPa in Fig. 4 are ~6%, which means that the spread in ensemble members is only 6% of the year-to-year variability in the streamfunction.

This is correct that the reanalysis data uncertainty is quite small, and more than one order of magnitude smaller than the interannual variability. We have added text to each section better describing the construction of the ERA5 ensemble and explaining why the variability is smaller than the inter-reanalysis and inter-annual variability. To summarize, the ERA5 reanalysis only reflects a portion of the uncertainty as the assimilation scheme itself is not varied, meaning that

structural uncertainty is not represented. In addition, we have changed the text to focus more on relative uncertainty, which is quite large when comparing SF and P-E to other metrics. The differences in uncertainty between metrics are as large as two orders of magnitude in each season, but the absolute uncertainty is usually much smaller than the interannual variability.

More importantly than showing this normalized STD for a field like the meridional streamfunction is showing what it means for the mean HC edge position and trends therein. From what I can glean from Figs. 1 and 2, the spread in SF edge trends is ~0.05° decade$^{-1}$ for the annual mean. Comparing to studies that have looked at multiple reanalyses (e.g., Fig. 4 of Davis and Rosenlof, 2012), SF edge trends span well over 1° decade$^{-1}$ among different reanalyses, and their uncertainties (for a single reanalysis, due to interannual variability) are ~0.5° decade$^{-1}$. If this were the only information available, I would say it shows that data-driven uncertainty is in fact not a substantial contributor to our understanding of historical tropical widening trends. The authors need to make a much more convincing argument as to why data-related uncertainty is substantial in the contexts of all of the other uncertainties related to different reanalyses, metrics, and interannual/natural variability.

This is a great point and based on this comment and others we have made substantial changes in the text to explain the importance of the results. In summary, the ensemble spread underestimates the uncertainty within a single reanalysis due to not including structural uncertainty, while the inter-reanalysis spread overestimates the uncertainty because of differences in assimilated observations, particularly when older reanalyses are used. As a result, the spread which we analyze is much smaller than the spread analyzed in previous studies, and is also much smaller than the inter-annual variability. However, the ERA5 ensemble allows for a systematic analysis of uncertainty within a single reanalysis product. Because the ERA5 spread is smaller than the true uncertainty, we focus on the differences in uncertainty between metrics. We have changed the text throughout the manuscript to more clearly explain the dataset and how the results compare to inter-reanalysis studies, as well as changed the results to focus more on the relative uncertainty.

Some more minor points are as follows:

Line 15: "data-driven uncertainty" – I don't agree that the ensemble member spread in tropical edge latitudes (or their trends) can simply be interpreted as "data-driven uncertainty" and meaningfully compared across metrics. The atmosphere may be very smooth/laminar/'predictable' in some places and much less so in others. For metrics based on fields that are 'predictable', one could easily get the same tropical width (or tropical width trend) among ensemble members even in an area with no data. Conversely, in highly chaotic/noisy/unpredictable regions, one might get a large spread in width/trend values even with relatively high quality/dense data to constrain the reanalysis. Ultimately, the only way to truly get at whether the differences in ensemble spread are "data-driven" vs. "atmosphere-driven" would be to compare the spread in an ensemble of free-running simulations to the spread in a (data-constrained) reanalysis. Without doing this, all you can say is that one metric is noisier than another. But one doesn't even really need reanalysis ensembles to demonstrate that, it has been shown before in a number of papers that have presented tropical width trends.

Yes, this is a good point and an unintended consequence of unclear language. We have changed the text to emphasize that the uncertainty is not directly related to observational errors, but instead the reanalysis error itself. We do speculate that poor observations are partly to blame for some of the reanalysis error as indicated by the improvements following improved observations, but aim to focus on the reanalysis data errors rather than the observation errors.

Line 17-18 "to date no study has quantified and compared the uncertainty in different HC metrics". This is simply incorrect. Numerous studies have both quantified and compared uncertainty in different HC metrics.

We have clarified that this is the first study to analyze the impact of reanalysis data uncertainty in a single reanalysis product.

Line 79: "several ensemble members" Just state how many there are.

Done

Lines 105 - 106: Please provide a reference describing the ensemble data assimilation.

Done

Lines 106 – 108: Please provide more details on the differences between the full ERA5 reanalysis and the ensemble version, as this is highly relevant to the interpretation of all of the results in this paper. For example, do both versions assimilate the same data? How different are the resolutions? Also, as all previous studies have used the full ERA5 reanalysis, it is critically important to also include the metrics and trends here for reference. Without doing this, we don't know whether there is some kind of bias in the lower resolution ensemble version of the reanalysis, or whether to expect that the results in this paper actually apply to the full reanalysis.

We have added text to clarify the differences between the products and explain that the results are similar. To summarize, both are quite similar, though the ensemble members have half the resolution and a data assimilation scheme with two inner loops rather than three. This does not result in noticeable differences in tropical extent estimates.

Line 110 – 112: Given that ERA5 resolution is ~30km (I'm not sure what the ensemble version is), reducing the data to 1 degree for your analysis seems like a really bad choice. Especially when you are trying to infer extremely small changes ~ 0.1 degree per decade. Some consideration needs to be paid to whether or not this degradation of the reanalysis resolution impacts your analysis.

This is an important consideration, and something that we looked into. The ERA5 ensemble members are 0.5-degree resolution, so we halved the resolution. Likely because TropD uses spline interpolation to calculate the tropical edge the decrease in resolution does not change the trends noticeably.

Line 111: "conservational" -> conservative

Done

Line 119: Please provide link/reference/version information for the python TropD code you are using.

All processing code is available on Zenodo with a link in the code availability section.

Line 126: "… poleward of the minimum … equatorward of the maximum … " this is correct for the NH but not the SH edge. Please clarify or make the language more general to apply to both hemispheres.

We have corrected this issue

Line 134: I believe Solomon et al. 2016 was the first to show this, and should be cited here in addition to Davis and Birner.

Great point, this was present in an earlier draft and is now included once again.

Solomon, A, L M Polvani, D W Waugh, and S M Davis. "Contrasting Upper and Lower Atmospheric Metrics of Tropical Expansion in the Southern Hemisphere." *Geophysical Research* 2016. https://doi.org/10.1002/2016GL070917.

Line 143-144: "[PSL is] …poorly correlated to P-E". From Waugh et al. 2018, the correlation between PSL (they call it SLP) and P-E is 0.32 in the NH and 0.69 in the SH. I would not call this correlation poor in the SH. Also, PSL is correlated with UAS in the SH with 0.98, so they are virtually interchangeable with one another. The authors should do a better job of making this distinction.

We have corrected the text to explain that PSL is moderately correlated with P-E and well correlated with SF. We also changed our review of PSL to explain that it is a reliable metric.

Line 153: "intermember average" (and many other points in the paper) - It is much more common to refer to this type of average as an ensemble average. I suggest changing this terminology throughout the paper.

Good point. We have changed this everywhere in the paper and in the figures.

Line 165-166 and Figure 1: I don't understand the need to do some kind of fit to 9 data points here, why not just show a simple histogram of the data? If the authors have a compelling reason to use these 'kernels', they should explain clearly, and also briefly explain Scott's rule.

We have added explanation of the KDE, why it is chosen, and how Scott's rule works. We choose the KDE rather than histogram as it allows for better visualization of the overlapping data and produces continuous estimates of the density of trends for each metric.

Line 177 (and possibly elsewhere in paper): "low" -> "small" The word low is appropriate for heights, not quantities.

This has been fixed throughout.

Lines 176-179: This sentence doesn't make sense. The authors state that the trends are a substantial downward revision but then that they are similar to Grise et al. It's not clear what the authors are trying to say here.

This sentence has been corrected to only present the results compared to Grise et al

Line 189-190: This goes to my point about "data-driven uncertainty" being a misnomer for describing the spread among reanalysis ensemble members. The authors state that over the SH, SF-based expansion is more robust (i.e., less uncertain). However, we know that reanalyses are less constrained by data in the SH. The fact that the "data-driven uncertainty" is smaller in the SH suggests that other factors are at play such as a stronger forced signal (e.g., due to ozone depletion) or less noise.

Yes, this is a good point. The reanalysis data error is the result of observation error and modeling error. We have changed the text to specify that we are focused on the error in the reanalysis data, rather than the observation errors. We have also corrected this section to indicate that both dynamics and reanalysis data error are causes of the Hadley cell extent uncertainty.

Figure 3: It is very hard to tell the difference in color between the UAS and PSL curves. Please consider using different colors.

We have tried many different color schemes and have found this to be the most appropriate. We believe that the issues in distinguishing between UAS and PSL in figure 3 are due to the overlapping lines.

Line 226: Are the reanalysis ensemble members really created by perturbing the model parameters?! Maybe the authors mean initial conditions here and not model physics, which is what I think of when I hear the word "model parameters". This is really important information for the reader to understand, as it has direct bearing on how the spread of the ensemble members is interpreted.

Yes, ERA5 includes perturbations in model tendencies which involves perturbing the model parameters. We have added some clarification to this.

Fig 6b: Is "SF uncertainty" just the ensemble STD of the SF in the vicinity of the edge (+- 2 deg) at 500 hPa? Or is it the ensemble average of the standard deviation within +-2 deg of the edge computed from each member? The caption doesn't specifically state what the statistic is, so this discussion is hard to evaluate.

This was initially unclear, and has been corrected to be more easily understood.

---

## Referee Report (RR1)

Review of

**Large Uncertainty in Observed Meridional Stream Function Tropical Expansion**
Baldassare et al.

**General**
This is my second review of this work, where the authors analyze the uncertainty in the SF metric for estimating the extent of the Hadley circulation (HC) based on ensemble members of the ERA5 reanalysis. As before, the key findings are a reduction over time of uncertainty, which the authors associate with better quality of the assimilated data, and that the SF metric has a relatively high uncertainty due to less observationally constrained upper wind and relatively weak meridional gradients near the zero-crossing latitude, which increase the uncertainty of the SF metric compared to other zero-crossing metrics. My main previous concerns were that (i) using only the ERA5 ensemble limits the generality of the results, (ii) the authors provided recommendations on the metrics which are not generally justified , (iii) the statistical implications of using only 9 members were not adequately discussed, and (iv) the motivation and results were lacking context. Of these comments, I can accept the revisions regarding the two first items, but I am not satisfied with the latter two items. As before, I think the analysis may merit publication. However, some critical issues remain. Detailed comments are provided below.

**General comments**
- On second reading, I suggest to rephrasing the title. Mainly, this is because "Meridional stream function tropical expansion" is obscure. First, the analysis applies to the edge of the HC as quantified by the meridional stream function. Second, tropical expansion is a tricky term since the tropical rain belt and tropical climate land area is actually narrowing (see recent paper by Adam et al. 2023, "reduced tropical climate land area under global warming"). For example, "Large uncertainty in observed estimates of Hadley cell expansion based on the meridional stream function" would be more appropriate (and consistent with the wording of the abstract).
- I still think that a qualitative and quantitative discussion of the limitations of using only 9 ensemble members is lacking.
- Figure 6a looks odd. How many data points are in this plot? I an able to identify 8 (4 seasons X 2). Of these 8, 3 lie along some line and 5 have essentially the same value of $\Delta$, and yet $R^2 = 0.97$? Clearly this is not a statistically robust result (for example, it would fail a cross validation test.). I would omit Fig. 6a and only keep 6b.

**Comments by line**
17-18 As I stated in  the previous review, I don't agree with this statement. Previous analyses have also considered the uncertainty of HD edge metrics. The current paper is likely the first to consider using the ERA5 ensemble members to estimate

the uncertainty. But other estimates exist, for example, like those now mentioned int he text (Davis and Rosenlof, 2012; the TropD paper; Davis and Birner, 2017; Waugh et al. 2018; Seviour et al. 2018; etc.). Specifically, using the ERA5 ensemble members to estimate 'observed' uncertainty is very much like estimating the differences across en ensemble of climate models, i.e., 'modeled uncertainty'. The authors should do a better job of delineating their work from existing works. This is done in the revised discussion, and to some degree in the introduction, but not in the revised abstract.

33    Instead of "tropical width" I would use "the extent of the tropical circulation" (or Hadley cell extent).

36    I suggest refraining from such recommendations.

59    Note the critical differences between this statement (which I agree with), and the one in the abstract (see above comment on lines 17-18)

69-72 As mentioned in the previous review, this is an odd choice for motivating the present analysis, as Chemke and Polvani find the models to generally agree on the extent trends. Surely you can find additional works to motivate the analysis. For example, the importance of having good estimates of observational uncertainty is demonstrated in State et al. (2020, BAMS). Specifically, in the early 2000's works like Johanson and Fu (2009) found large discrepancies between modeled and observed trends, which sent many researchers rushing to identify the errors in the models, only to realize a decade later than the error was in the observed trends.

188   I believe kernel density estimates require some assumptions. Please provide a reference and provide the reader with all the information required to exactly reproduce your results.

241   remove "sometimes"

282   In the revised text you added that the change from 6% to 4% is not statistically significant. According to what criteria?

---

## Author Response (AR2)

Response to Review of

**Large Uncertainty in Observed Meridional Stream Function Tropical Expansion**

Baldassare et al.

**General**

This is my second review of this work, where the authors analyze the uncertainty in the SF metric for estimating the extent of the Hadley circulation (HC) based on ensemble members of the ERA5 reanalysis. As before, the key findings are a reduction over time of uncertainty, which the authors associate with better quality of the assimilated data, and that the SF metric has a relatively high uncertainty due to less observationally constrained upper wind and relatively weak meridional gradients near the zero-crossing latitude, which increase the uncertainty of the SF metric compared to other zero-crossing metrics. My main previous concerns were that (i) using only the ERA5 ensemble limits the generality of the results, (ii) the authors provided recommendations on the metrics which are not generally justified, (iii) the statistical implications of using only 9 members were not adequately discussed, and (iv) the motivation and results were lacking context. Of these comments, I can accept the revisions regarding the two first items, but I am not satisfied with the latter two items. As before, I think the analysis may merit publication. However, some critical issues remain. Detailed comments are provided below.

Thank you for taking the time to review our manuscript for the second time. We have made further revisions to better explain the statistical implications of using only 9 ensemble members and to better contextualize the results within the existing literature.

**General comments**

- On second reading, I suggest to rephrasing the title. Mainly, this is because "Meridional stream function tropical expansion" is obscure. First, the analysis applies to the edge of the HC as quantified by the meridional stream function. Second, tropical expansion is a tricky term since the tropical rain belt and tropical climate land area is actually narrowing (see recent paper by Adam et al. 2023, "reduced tropical climate land area under global warming"). For example, "Large uncertainty in observed estimates of Hadley cell expansion based on the meridional stream function" would be more appropriate (and consistent with the wording of the abstract).

We have changed the title to be both less obscure and more accurate.

- I still think that a qualitative and quantitative discussion of the limitations of using only 9 ensemble members is lacking.

We have added two sentences to the data subsection of the methodology explaining the uncertainty resulting from the small number of ensemble members including how this does not impact the comparison between SF and P-E and the other metrics which are typically orders of magnitude apart on uncertainty. We have also added an explanation of the value of the fractional uncertainty.

In addition, we have revised the text on line 282 to explain how the fractional uncertainty impacts the statistical significance.

- Figure 6a looks odd. How many data points are in this plot? I an able to identify 8 (4 seasons X 2). Of these 8, 3 lie along some line and 5 have essentially the same value of $\Delta$, and yet $R^2 = 0.97$? Clearly this is not a statistically robust result (for example, it would fail a cross validation test.). I would omit Fig. 6a and only keep 6b.

We have removed the best fit line from Fig. 6a and added labels marking the season for each data point, which is now the focus of the discussion. The section of the text describing Fig. 6a has been rewritten to discuss only the large delta values and uncertainty present in the NH in JJA and SON. Because the delta impact seems to only be present in the NH in these seasons, we believe this better explains the important results without making a more general claim about a linear relationship between delta and uncertainty which we do not believe exists.

**Comments by line**

17-18  As I stated in the previous review, I don't agree with this statement. Previous analyses have also considered the uncertainty of HD edge metrics. The current paper is likely the first to consider using the ERA5 ensemble members to estimate the uncertainty. But other estimates exist, for example, like those now mentioned int he text (Davis and Rosenlof, 2012; the TropD paper; Davis and Birner, 2017; Waugh et al. 2018; Seviour et al. 2018; etc.). Specifically, using the ERA5 ensemble members to estimate 'observed' uncertainty is very much like estimating the differences across en ensemble of climate models, i.e., 'modeled uncertainty'. The authors should do a better job of delineating their work from existing works. This is done in the revised discussion, and to some degree in the introduction, but not in the revised abstract.

This has now been corrected to specify that this is the first study to consider the impact of reanalysis data error.

33  Instead of "tropical width" I would use "the extent of the tropical circulation" (or Hadley cell extent).

This has been changed.

36  I suggest refraining from such recommendations.

This has been removed, and in its place a general discussion of the uncertainty differences between metrics.

59  Note the critical differences between this statement (which I agree with), and the one in the abstract (see above comment on lines 17-18)

69-72  As mentioned in the previous review, this is an odd choice for motivating the present analysis, as Chemke and Polvani find the models to generally agree on the extent trends. Surely you can find additional works to motivate the analysis. For example, the importance of having good estimates of observational uncertainty is demonstrated in State et al. (2020, BAMS). Specifically, in the early 2000's works like Johanson and Fu (2009) found large discrepancies between modeled and observed trends, which sent many researchers rushing

to identify the errors in the models, only to realize a decade later than the error was in the observed trends.

The sentence referring to Chemke and Polvani has been removed. In addition, the sentence referencing Staten et al. (2020) in the introduction has been altered to explain that previous work has found a connection between reanalysis data errors and questionable expansion rates.

188    I believe kernel density estimates require some assumptions. Please provide a reference and provide the reader with all the information required to exactly reproduce your results.

A subsection describing the kernel density estimate has been added to the Methods. This includes a reference to Silverman (2018), the modern edition of Silverman's 1986 book describing the kernel density estimate, a brief description of the method, and a description of the code used to implement the kernel density estimate. Text has been removed from the Results, and instead replaced with a link to the Methods.

241    remove "sometimes"

Done

282    In the revised text you added that the change from 6% to 4% is not statistically significant. According to what criteria?

Text has been added to the Methods describing the fractional uncertainty and the issues with having only 9 ensemble members. This line has been edited to explain that this change is not statistically significant at the 95% confidence level due to the 25% fractional uncertainty.